# The Long and Short of Biodiversity: Cumulative Diversity and Its Drivers

**Matthew Hammond \*** and **Jurek Kolasa**

Department of Biology, McMaster University, 1280 Main St. West, Hamilton, Ontario, L8S 4K1, Canada; kolasa@mcmaster.ca
\* Correspondence: mhammond@gmx.com

**Abstract:** Long-term or cumulative diversity is the biodiversity that accumulates at a site over many generations of community members. Cumulative diversity is likely important to the intrinsic and functional value of ecosystems given the legacies left behind by many species. While its components—average short-term diversity (alpha) and temporal turnover (beta)—have been extensively studied, cumulative diversity itself has not. We therefore examined the environmental and community drivers of cumulative diversity with a novel hierarchical diversity partition. This partition breaks cumulative diversity into short-term, turnover, richness, and evenness components. We applied this framework to 49 tropical rock pool communities, censused over tens to hundreds of organism generations. Results uncovered two environmental regimes that differentially impacted the richness and evenness components of cumulative diversity: Occasional drying events mainly limited richness and reset communities, while less severe physicochemical variations reduced the evenness of communities. These causal pathways amount to differential controls on cumulative diversity; controls that can oppose each other to buffer diversity against change as well as create unexpected trade-offs for managers. We conclude that maintaining diversity at longer timescales requires new analytical tools and an expanded view that can account for its complexity.

**Keywords:** diversity partitioning; temporal; alpha; beta; cumulative diversity; turnover; environmental variation

## 1. Introduction

Managing the variety of life on Earth is complicated by the fact that diversity patterns change with the timescale of observation. For instance, species accumulate at a site over time such that short-term and long-term inventories can differ substantially [1]. As a result, a site may have low diversity at any given census but high diversity over longer time periods. This scale-dependency of diversity has led to earnest pleas for considering how the passage of time affects the census and interpretation of biodiversity patterns [2].

Most diversity investigations apply a fine temporal grain, censusing species over short sampling periods, such as within a generation or two of the average community member. This is short-term or alpha diversity. Over short sampling periods, censused species are largely contemporaneous and their diversity reflects the variety of organisms that can inhabit a site, interact and do "ecological work" at the time. Short-term diversity is therefore the main yardstick for assessing a site's capacity to support communities (e.g., amount of local diversity) and ecosystem processes (e.g., diversity of functional species). It is important to note that even though short-term diversity can be tracked over long periods of time (e.g., tens of thousands of years [3]), the fine grain of analysis reveals only trends in the diversity of contemporaneous species. It does not uncover the full biological diversity that influences a site over longer timespans.

A different picture can emerge when biodiversity is censused over many generations of the target organisms. We call this quantity long-term or cumulative diversity after [1]. It is obtained by repeated sampling and aggregating short-term data for a coarser-grained inventory of species that have accumulated at a site with time [4]. Despite being understudied, evidence is growing that long-term diversity may be an important ecological quantity. High biodiversity value in some freshwater ponds, for example, emerges over time from a continual turnover of species [5].

Long-term diversity may also shape local ecosystems functionally in ways that contemporaneous species do not. For example, successional and invasive species leave physical and community legacies that extend well beyond their tenure at the site [6,7]. Historical overgrazing by deer can severely perturb forest communities [8]. Periodic insect outbreaks (e.g., cicadas or locusts) can modulate above- and below-ground ecosystem processes with considerable time lags [9]. Rare and infrequent coral reef species can similarly provide vital ecosystem functions despite not being permanent members of the community [10,11]. There is also research that shows that species diversity at million-year timescales stabilizes coral reef functioning [12]. It is possible then that levels of time-accumulated diversity are as significant, both intrinsically and functionally, as short-term diversity.

While research unravels the ecological significance of long-term diversity, we focus on its statistical and mechanistic drivers. Statistically speaking, long-term diversity arises from shorter-term diversity patterns. Whittaker's [13] classical partition applied in the temporal dimension, for instance, decomposes long-term or cumulative diversity as a gamma diversity ($\gamma_D$) into the average diversity of a censused community ($\alpha_D$)—a measure of short-term diversity—and the temporal turnover of that diversity ($\beta_D$):

$$\gamma_D = \alpha_D \beta_D \tag{1}$$

While the alpha and beta components of cumulative diversity have been extensively studied using various methodologies [13–15], cumulative diversity itself has not. Similarly, while the species–time relationship has effectively addressed the patterns and theoretical mechanisms of species accumulation [16,17], the factors that promote and sustain long-term diversity in natural systems are elusive.

Biologically, several non-exclusive mechanisms can account for levels of long-term diversity. The most generic and neutral of these is dispersal-driven metacommunity dynamics: Species colonize a site from the surrounding metacommunity and spur the detection of new species and growth of long-term diversity [17,18]. Phenomenologically, this growth tends to occur from occasional, low abundance species adding to a persistent core of abundant species [2,19,20]. Where meaningful environmental gradients exist, local conditions filter the regional species pool to dictate community composition and structure [21]. Local environmental conditions are often taken as the site average. However, because environments are dynamic, fluctuations of physicochemical variables—the temporal dimension of the environment—may strongly influence the build-up and maintenance of diversity [22]. On the organismal supply side, members of the regional species pool may be specialized (e.g., life history traits) to thrive in specific, even variable environments and so boost local diversity [23–25].

Here, as an illustration of methodology and mechanisms at play, we uncover the community and environmental drivers of long-term diversity in a biodiverse rock pool system using a new hierarchical partition that breaks long-term diversity into alpha, beta, richness and evenness components. The advantage of this approach lies in detecting the root causes of variation in long-term diversity, ranging from impacts on richness to evenness. We use the method to answer three questions critical to understanding and preserving long-term diversity in landscapes:

(1)  What factors support cumulative diversity?
(2)  Do richness and evenness components contribute differently to cumulative diversity in the face of environmental change?
(3)  How do multiple paths of causation combine to determine cumulative diversity?

## 2. Materials and Methods

### 2.1. Hierarchical Partitioning of Cumulative Diversity

Cumulative diversity refers to the diversity of a temporally-aggregated sample, which is a temporal $\gamma$ diversity. Measures recently proposed by [26,27] provide a basis for partitioning such cumulative diversity into richness and evenness components. To do so, we applied Tuomisto's partition of diversity into independent components: Diversity = Species richness x Evenness, where diversity is measured as effective species number (i.e., Hill's number) and evenness is the ratio of diversity to species richness [27]. Applying this partition to the temporal domain, we decomposed cumulative diversity ($\gamma_D$) into components of cumulative richness ($\gamma_R$) and cumulative evenness ($\gamma_E$):

$$\gamma_D = \gamma_S \gamma_E \tag{2}$$

where $\gamma_S$ and $\gamma_E$ are the richness and evenness of temporally-aggregated abundance data, respectively.

From [26], we noted that gamma measures like $\gamma_S$ and $\gamma_E$ can each be partitioned into alpha and beta components of the forms: $\gamma_S = \alpha_S \beta_S$ and $\gamma_E = \alpha_E \beta_E$. Substituting these identities into Equation (2) adds a lower hierarchical level of richness and evenness in alpha and beta components to the definition of cumulative diversity:

$$\gamma_D = \alpha_S \beta_S \alpha_E \beta_E \tag{3}$$

where (1) $\alpha_S$ is the average richness of a sampled assemblage, (2) $\beta_S$ is the compositional turnover of assemblages in time, (3) $\alpha_E$ is the average evenness of a sampled assemblage and (4) $\beta_E$ is the turnover of evenness in time, which is the result of changing species dominance. We calculated these components for each rock pool as diversity indices of order $q = 2$ (i.e., the Hill number exponent for number of effective species) according to formulae from [26,27]. The formulae are reported in Table S1.

### 2.2. Rock Pool System and Community Sampling

Our study system was a set of 49 coastal rock pools near Discovery Bay Marine Laboratory, University of the West Indies (Figure 1), on the northern coast of Jamaica (18°28′ N, 77°25′ W). Pools, which have volumes ranging from 0.5 to 78.4 L, lie on a 25 m radius section of fossil reef within 10m of the ocean. Pools are refilled by rain, ocean spray and, for a few, by spring tides. The metacommunity hosts 79 invertebrate species which disperse as propagules by wind, ocean spray, animal vectors and overflow after heavy rains [28]. Species are divided among: Ostracods (21 species), copepods (nine species), cladocerans (three species), worms (16 species), aquatic insects (22 species) and other invertebrates (eight species).

We sampled invertebrate communities over 14 annual surveys (1989–2003) by withdrawing 0.5 L of water after stirring the pool to dislodge organisms from rock walls and homogenize the contents. We filtered each sample through 63 μm mesh to isolate invertebrates (excluding rotifers and gastrotrichs which pass through) prior to preservation in 50% ethanol and counting by microscope. Environmental variables of temperature, salinity, dissolved oxygen, pH and chlorophyll-a concentration (a proxy for a pool's biological productivity) were measured with multiprobe sondes (DataSonde, Yellow Springs Instruments, Yellow Springs, Ohio, USA or Hydrolab, Austin, Texas, USA). These data were available for 8–11 of the survey years, except for chlorophyll-a, which was measured on six annual surveys. Occasional drying of some pools during periods of limited rain precluded their sampling. These occasions were recorded as drying events and entered as blanks (unsampled) in the community data.

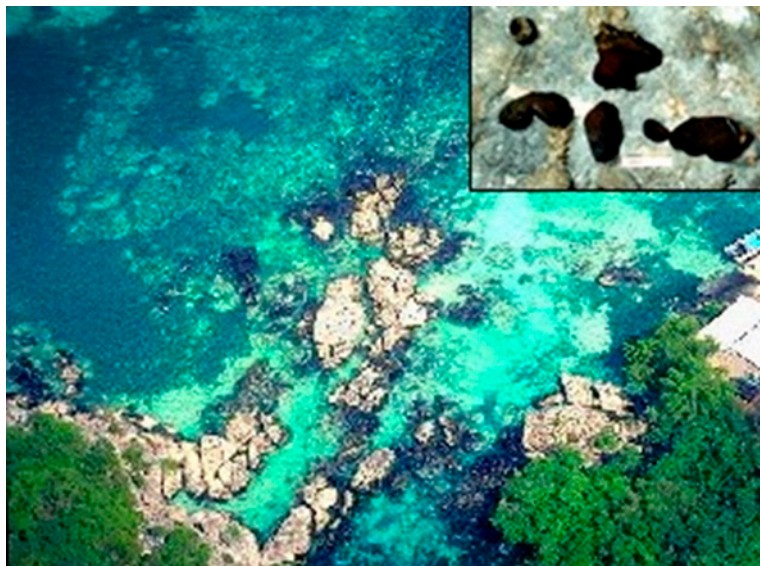

**Figure 1.** Aerial view of the rock pool system at Discovery Bay, Jamaica (Discovery Bay Marine Lab, University of West Indies). Inset: Close-up of a subset of rock pools.

### 2.3. Environmental and Community Factors

To evaluate the extent to which cumulative diversity depends on environmental variation, we used principal components analysis (PCA) to summarize the prevailing regimes of environmental variation in the rock pool system. Previous work in the system [29,30] and preliminary regressions (data not shown) indicate that rock pool diversity patterns are driven more by variation in environmental conditions than by mean conditions. We therefore included as variables coefficients of variation of temperature, salinity, dissolved oxygen, pH and chlorophyll-a, as well as the number of recorded drying events. Principal components were judged significant if they explained more than 25% of variance and came before the "elbow" of the scree plot.

To assess the degree to which cumulative diversity depends on the presence of specialists, we calculated the mean habitat specialization of species in a rock pool. We measured a species' habitat specialization as the inverse of Levin's niche breadth [31] which, when applied in space, measures a species abundance across different habitats [29], as follows:

$$B_j^{-1} = \sum_i^N p_{ij}^2 \tag{4}$$

where $B_j^{-1}$ is the niche (habitat) specialization of species $j$ and $p_{ij}$ is the proportion of individuals of species $j$ found in pool $i$ over all sampling years. This approach defines each pool as a distinct "resource state" a priori and not based on true environmental differences between pools. It therefore provides a more general measure of concentration in a heterogeneous landscape and hence exposure to a limited range of local conditions. Equivalent to Simpson's concentration index [32], $B_j^{-1}$ is high for species that are concentrated in few pools and low for widespread species.

We noted that a species' spatial distribution is not independent of its abundance [33] and so our habitat specialization measure was confounded with abundance at extremely low levels: A species with one individual, for instance, necessarily occurred in only pool and registered as specialized. For this reason, we entered mean local abundance of species in a pool as a covariate in statistical models. However, while local abundance was intended as a covariate, it also has biological meaning as a driver of cumulative diversity. A more diverse community, for instance, may have lower mean abundance if low abundance, occasional species are added to a core set of high abundance species over time [2,19,20]. Data simulations confirmed that a negative relationship between the average

abundance of species in a pool and cumulative diversity or richness arose when diversity or richness increased from the accumulation of rare species over time (Figures S1–S3). Alternatively, a negative abundance–diversity relationship could mean that added species compete for resources and reduce average population size [34]. Since this relationship is rarely reported, we interpreted a negative regression slope as evidence for the accumulation of rare species. A positive slope, on the other hand, can indicate diversity increases from the accumulation of abundant species (Figure S2) or, alternatively, higher diversity in more productive systems (see meta-analysis by [35]).

We performed multiple regressions in Statistica v.10 (StatSoft Inc., Tulsa, Oklahoma, USA) to test responses of cumulative diversity and its components to environmental variation, habitat specialization and local abundance of community members. Data were log- or square-root-transformed when needed to meet parametric assumptions. We assessed the importance of individual factors in multiple regressions using their semi-partial correlations which estimate their explanatory power [36].

## 3. Results

Cumulative diversity ($\gamma_D$) was 1.6-fold higher than alpha diversity ($\alpha_D$) on average. Put another way, 35% $\pm$ 16 (SD) of effective species found in the long-term species pool were missing from an average assemblage. This figure grew to 67.9% $\pm$ 7.1 for cumulative species richness. Categorizing individual species according to their extinction/colonization patterns showed that the cumulative species pool was dominated by recurring species (Table 1): Cyclic species that recurred intermittently contributed most (53%) to the cumulative pool while permanent species (found in all years) were rare (3.3%). Transient species also contributed substantially (34.5%) to cumulative richness, while directional losses from and additions to the initial community were minor (<10% each). Percentage of cyclic species was positively associated with alpha richness ($r = 0.63$, $p < 0.001$), while more new additions and transient species led to higher compositional turnover or $\beta_S$ ($r = 0.51$, $p < 0.001$; $r = 0.49$, $p < 0.001$).

**Table 1.** Composition of the cumulative species pool in terms of permanent, cyclic, transient, lost and newly-added species as defined below. For categorizing, a species was considered extinct if it was absent after being present in the previous year.

| Species Type | Mean % | Standard Deviation | Extinction/Colonization Pattern |
|---|---|---|---|
| Permanent | 3.3 | 4.8 | Present in all annual samples |
| Cyclic | 53.0 | 13.0 | Recurs intermittently at least once |
| Transient | 34.5 | 10.7 | Not initially present, colonizes and goes extinct, does not recur |
| Lost from site | 3.2 | 5.0 | Initially present, goes extinct, does not recur |
| New addition | 6.1 | 7.3 | Not initially present, recurs every year after colonization |

We identified three factors that limited cumulative diversity of a rock pool ($R^2 = 0.59$, F = 15.52, $p < 0.0001$; Tables S2 and S3): Two types of environmental variation that together explained >40% of variance (based on squared semi-partial correlations) and the mean abundance of local populations, which explained a further approximately 10% of variance.

Environmental variation (EV) took two forms, captured by the first two axes of a PCA and together explaining 57.2% of variance in the environmental dataset. The first axis (EV 1) summarized physicochemical variation in the form of fluctuations in temperature, pH and productivity (Table 2). This variation was most apparent in shallow pools ($R^2 = 0.54$, $F_{6,42} = 8.23$, $p < 0.0001$) which were warmer on average ($r = 0.57$, $p < 0.001$) and had the most variable productivity ($r = 0.28$, $p = 0.048$) and pH ($r = 0.04$, $p = 0.004$). This pattern was consistent with environmental variability arising from ecosystem responses to extrinsic forcing: Solar heating promoting thermal spikes that lead to intermittent decomposition, algal blooms and attendant pH fluctuations. The second axis of environmental variation (EV 2) summarized the drying–rewetting cycles of the rock pool system that desiccate pools with the highest surface areas and lowest water volumes ($R^2 = 0.37$, $F_{6,42} = 4.09$, $p = 0.003$). Factor loadings (Table 2) show that cycles were associated with variable salinity as pools

evaporated and refilled as well as more stable oxygen levels, likely owing to more atmospheric equilibration across the high surface area of affected pools.

**Table 2.** Sets of environmental drivers in system of Jamaican rock pools, summarized by principal components analysis (PCA). Variance explained by PCA axis in parentheses, significant factor loadings in bold.

| Variable | EV 1 (28.1%) | EV 2 (29.1%) |
|---|---|---|
| Temperature variability | **0.72** | 0.09 |
| Salinity variability | −0.44 | **0.71** |
| Dissolved $O_2$ variability | −0.15 | **−0.61** |
| pH variability | **0.66** | 0.29 |
| Chlorophyll *a* variability | **0.67** | −0.46 |
| Number of sampling days dry | 0.37 | **0.72** |

Diversity partitioning allowed us to identify the main pathways by which environmental and community factors impacted cumulative diversity. The Whittaker partition highlighted a single factor—physicochemical variation—that curtailed cumulative diversity by reducing both alpha ($\alpha_D$) and beta diversity ($\beta_D$; Figure 2). Because environmental variation reduced beta diversity to very low levels (Figure 2b), values of cumulative and assemblage diversity converged in the most variable pools (Figure 2a). However, we did not detect any effect of local abundance or pool drying on $\alpha_D$ or $\beta_D$ even though they both reduced cumulative diversity in regressions (Figure 3a).

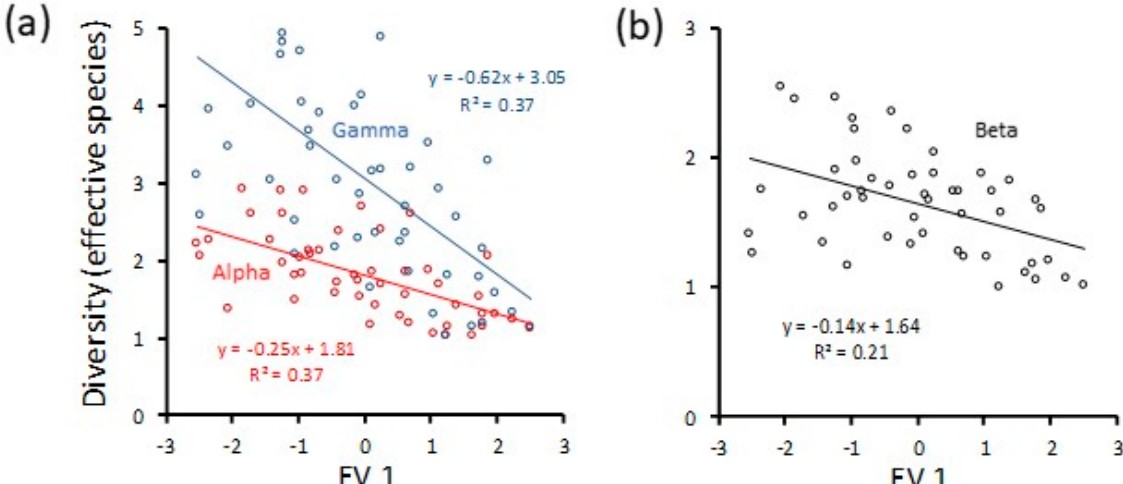

**Figure 2.** Environmental drivers of diversity in Jamaican rock pools according to the Whittaker partition. (**a**) Responses of cumulative diversity ($\gamma_D$) and assemblage diversity ($\alpha_D$) to a composite of fluctuating physicochemical variables, environmental variation 1 (EV1). Slopes differ significantly ($t = 3.08$, $p = 0.003$). Cumulative and assemblage diversity become similar at high values of EV1 because (**b**) beta diversity ($\beta_D$) declines and approaches one along the environmental gradient.

Our hierarchical partition, based on measures proposed by [26,27], added further detail of the factors influencing cumulative diversity. Results showed multiple paths by which drivers affected the richness and evenness components of cumulative diversity, as well as their alpha and beta components (Figure 3b). In a richness path (Figure 3b, left branch), cumulative diversity depended on impacts to species richness components. Pool drying caused declines in cumulative species richness ($\gamma_S$) by negatively affecting alpha richness ($\alpha_S$) lower in the hierarchy. In contrast, physicochemical variation had a smaller effect on cumulative richness because of countervailing impacts: It reduced the average community richness ($\alpha_S$) but increased compositional turnover ($\beta_S$). This partial compensation

blunted the negative effects of environmental variation on cumulative richness by an estimated 67% (Figure S4a).

In an evenness path (Figure 3b, right branch) cumulative diversity depended on impacts to evenness components. The evenness path was governed mainly by physicochemical variation which reduced cumulative evenness ($\gamma_E$) by limiting the turnover of evenness ($\beta_E$). The impact of drying on cumulative evenness was blunted by 125% by another partial compensation: Drying increased the evenness of a community ($\alpha_E$) but reduced evenness turnover ($\beta_E$; Figure S4a), such that it was not a significant predictor of cumulative evenness.

Other factors were less influential but nonetheless highlighted the non-environmental drivers of cumulative diversity. A negative relationship between cumulative richness and local abundance was consistent with species count increasing through the addition of rare species. Habitat specialization had various effects and was associated with the occurrence of purely marine species. This marine specialization was indicated by increases in mean specialization with pool salinity and decreases in specialization with the range of salinity experienced in a pool ($R^2 = 0.20$, $F_{2,46} = 5.78$, $p < 0.0001$; Table S4). Specialization increased the average richness of a community which led to higher cumulative richness. However, it also lowered cumulative evenness by reducing average community evenness—yet another countervailing effect reducing the impact of specialization on cumulative diversity by an estimated 65% (Figure S5).

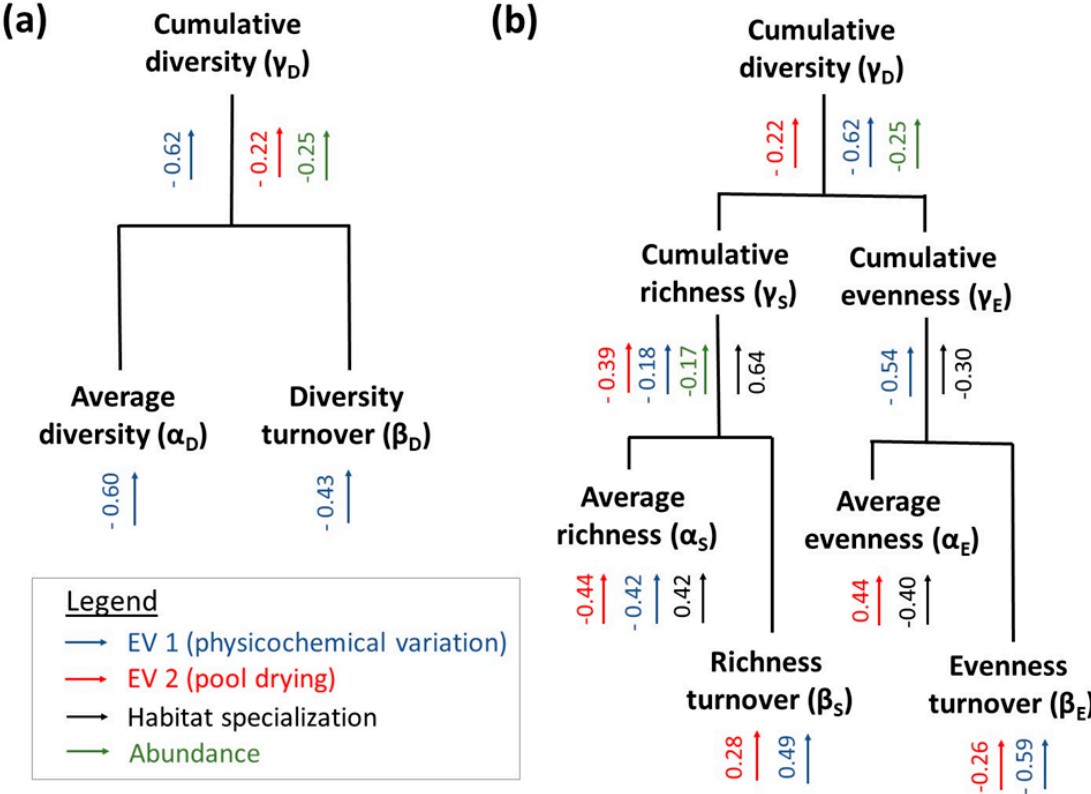

**Figure 3.** Visual summary of multiple regression results of environmental and community factors predicting (**a**) the temporal alpha and beta components of cumulative diversity using the Whittaker partition and (**b**) the richness and evenness alpha and beta components from a more nuanced diversity partition (Equation (3)). Coloured numbers report the semi-partial correlations of factors in predicting diversity components. Because components are hierarchical, positive correlations at lower levels increase the basis for long-term diversity unless cancelled by effects on other components.

## 4. Discussion

A site only houses a fraction of the biodiversity and ecosystem functions present in a landscape [37]. The temporal analog of this—that not all species and essential functions are contemporaneous at a site—may also be true [38]. Diversity that accrues in time thus emerges as a potentially critical, yet underappreciated, aspect of biodiversity. We applied a new hierarchical framework to uncover the key determinants of diversity accumulated over tens to hundreds of organism generations. The existence of multiple pathways through which long-term diversity builds up highlights two messages: First, that long-term diversity is under significant control by environmental variability and, second, that the nature of this control may differ for richness and evenness, with implications for conserving diversity over long timescales.

### 4.1. Environmental Control of Cumulative Diversity

Cumulative diversity is governed by relatively few factors and is highly sensitive to environmental variation. Just three factors captured more than half the variation in cumulative diversity, while four factors explained 77% of cumulative richness in the rock pool system. Environmental variation is commonly reported to limit diversity and richness over short timescales (i.e., $\alpha_S$ and diversity $\alpha_D$ [39]). However, here we show a long-term impact in which cumulative diversity is doubly sensitive to environment through the erosion of both temporal alpha and beta diversity (Figure 2b).

Loss of diversity turnover is particularly striking given that community change is the expectation when many generations are tracked [40]. It appears, however, that strong environmental variation can virtually halt turnover (for a counterexample, see [41]) and thus depress levels of long-term diversity. Indeed, long-term diversity was no larger than short-term diversity in the most dynamic environments (Figure 2a). The erosion of the long-term diversity likely comes as environmentally-forced communities become simplified assemblages of the most tolerant and persistent species [42,43]—in essence, a temporal form of biotic homogenization [44]. Moreover, our results indicate that the exclusion of new community members and transient species (see Table 1) may be particularly damaging to turnover, while the loss of cyclically recurring species may limit alpha richness. The broader implication is that global trends towards more variable environments may do much to curb cumulative diversity as well as its support for stable ecosystem functioning [38,45].

Community and metacommunity properties also determined cumulative diversity, but were far less influential than environmental variation (Figure 3). A negative local abundance–diversity relationship (Figure 3b), for instance, suggested that cumulative diversity built up through the metacommunity dynamic of occasional species (infrequent, low abundance) adding to a core community [2,19]. Further support for this comes from the predominance of impermanent species (cyclic and transient) in the long-term species pool. Prior work in the same system suggests that multiple mechanisms—including species sorting, mass effects, patch and neutral dynamics—may act in concert to generate this metacommunity dynamic [30]. Maintaining cumulative diversity in such systems is likely to depend on both local- and landscape-scale conservation strategies. At the local scale, permanent and cyclic species may be regarded as the recurring diversity particular to a site. As such, populations of permanent community members might be managed directly over time while optimal environmental conditions are maintained for the persistence of cyclic species. High numbers of transient and cyclic species also make the case for landscape-scale conservation that preserves the source populations and immigration routes of occasional species, which are thought to be especially migratory [46].

Communities of habitat specialists also tended towards greater cumulative richness (but not cumulative diversity), underscoring ecological niche specialization as an important correlate of diversity [29,47]. From a management standpoint, these findings imply the need to prioritize specialist-associated habitats—marine in this case—and protect resident specialists against the extinction risks of narrow niche breadths and small geographical ranges ([48] but see [49]). We note that cumulative richness and diversity can also increase when a greater sampling effort over time

captures difficult-to-detect species [17,18]. This sampling effect should not distort our regression results since each pool had the same number of samples with, presumably, the same sampling error.

### 4.2. Differential Pathways to Cumulative Diversity

Twin and uncorrelated environmental regimes eroded cumulative diversity through effects on cumulative richness and evenness (the middle hierarchical level of Figure 3b). Two causal pathways, which would not be detected by classical diversity partitions, were apparent. A richness path saw cumulative richness lowered by infrequent drying events (EV 2) and physicochemical variation (EV 1) which mainly limited the size of communities ($\alpha_S$), in agreement with [50]. An evenness path saw cumulative evenness decline mainly through physicochemical variation that prevented changes in dominance ($\beta_E$). Mechanistically, this dominance-switching—if not prevented—would promote long-term evenness by limiting the sustained dominance of any one species (see [51,52] for importance in paleobiology). These pathways constitute a dual control on levels of cumulative diversity. Findings, in turn, imply that different management strategies may be required for managing the various richness and evenness components of diversity over larger timescales. Alpha richness and evenness, for instance, may be well-served by existing management techniques that seek to maintain community size and structure. However, strategies must also be developed that enable compositional and dominance turnover, such as the maintenance of cyclic environmental regimes.

Despite the negative impacts of environmental variation, cumulative diversity was buffered against the worst effects by many compensations within and among pathways (lower hierarchical levels of Figure 3b). These compensations occurred when a negative impact on one component of cumulative diversity was offset by a positive impact on another component (cf. [39]). Within the richness path, environmental variation increased compositional turnover ($\beta_S$) at the same time it reduced species richness ($\alpha_S$). This compensation (plus a similar one within the evenness path) suggests that diversity losses that are catastrophic at short timescales may be inconsequential over longer time periods if they are balanced by gains due to turnover. We estimate that this mechanism stabilized rock pool cumulative diversity by more than 50%. Similar compensations existed between paths, such as habitat specialization or environmental impacts that decreased evenness but increased richness. Given this ubiquity, we suggest that the existence of multiple causal pathways will be important for stabilizing cumulative diversity in the face of stressors.

While they may buffer cumulative diversity against declines, multiple pathways could also pose difficult trade-offs for ecosystem managers. For instance, should a stable environment be encouraged if it promotes richness but limits turnover? Or, should specialized communities with high richness be sought even if, as here, they come with a cost of unevenness? Optimizing management in the face of such trade-offs is a challenge for the future. For now, such questions may force ecologists to be clearer about exactly which features of biodiversity—whether richness, evenness, community size or turnover—they wish to conserve when they protect it.

We believe our findings are generalizable even though not every community will be structured by the same factors as Jamaican rock pools. Multiple causations of temporal turnover simultaneously driven by physical, ecological and geographical factors has been reported [53]. Human impacts add a further key causal factor of community patterning [54]. In the tangle of drivers, features like compensations between pathways should be widespread since they are emergent properties of aggregate systems. Indeed, these pathways add a new example of compensations known to stabilize ecological systems, from alternative energy pathways in food webs [55] to differential responses of species to fluctuating environments [56]. Understanding and navigating this kind of complexity in the build-up of biodiversity is therefore a key management challenge in a time of global environmental change. We suggest meeting this complexity head-on with (1) new statistical tools capable of dissecting cumulative diversity and detecting dynamic features like compensations and (2) management strategies that take a longer-term view and account for multiple pathways of environmental variation.

**Supplementary Materials:** The following are available online at http://www.mdpi.com/1424-2818/11/3/41/s1, Figure S1: Simulated relationships between mean local abundance of species and (**a**) cumulative diversity, (**b**) cumulative richness and (**c**) cumulative evenness when species accumulate in order from high to low abundance, Figure S2: Simulated relationships between mean local abundance of species and (**a**) cumulative diversity, (**b**) cumulative richness and (**c**) cumulative evenness when species accumulate from low to high abundance Figure S3: Simulated relationships between mean local abundance of species and (**a**) cumulative diversity, (**b**) cumulative richness and (**c**) cumulative evenness when species accumulate in random order of abundance, Figure S4: Opposing effects of environmental variation on components (**a**) within the richness path and (**b**) within the evenness path that cancel out and buffer impacts at higher hierarchical levels (on cumulative richness and cumulative evenness, respectively), Figure S5: Opposing effects of environmental variation on cumulative richness and cumulative evenness that cancel out and buffer environmental impacts on cumulative diversity, Table S1: Diversity measures used in Whittaker partition and the new hierarchical partition with measures from Tuomisto [26,27], Table S2: Results of multiple regression models predicting diversity measures from Whittaker's partition (Equation (3)) as a function of environmental PCA axes (Physicochemical variation, EV 1 and pool drying, EV 2), mean habitat specialization and mean local abundance, Table S3: Results of multiple regression models predicting diversity measures from a new hierarchical partition (Equation (3)) as a function of environmental PCA axes (Physicochemical variation, EV 1 and pool drying, EV 2), mean habitat specialization and mean local abundance, Table S4: Results of multiple regression predicting mean habitat specialization as a function of mean salinity and salinity range of a pool. (Reference [57] is cited in supplementary materials).

**Author Contributions:** Conceptualization, M.H. and J.K.; Data curation, J.K.; Formal analysis, M.H.; Funding acquisition, J.K.; Methodology, M.H.; Project administration, J.K.; Writing—original draft, M.H.; Writing—review & editing, J.K.

**Funding:** This study was supported by funding from the Natural Sciences and Engineering Council of Canada, grant # RFMAC-10531314.

**Acknowledgments:** M.H. and J.K. gratefully acknowledge several years of support from the Natural Sciences and Engineering Research Council of Canada. We thank the Discovery Bay Marine Lab, University of West Indies for logistical support and many past graduate students and volunteers for aiding data collection.

**Conflicts of Interest:** The authors declare no conflict of interest.

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
