# Peer review of "The Long and Short of Biodiversity: Cumulative Diversity and Its Drivers"

_diversity, doi:10.3390/d11030041_

Round 1

Reviewer 1 Report

Review of Hammond and Kolasa, The long and short of biodiversity: Cumulative diversity and its drivers. Submitted to Diversity (manuscript 460132).

The authors introduce an approach to the study of cumulative diversity and apply this approach to tropical rock pool communities. The general concepts are interesting and potentially useful. The study is succinct and well written.

Main comments:

1. It is not clear to me whether the authors believe diversity accumulates due to directional or cyclical changes in species assemblages (or both). This distinction seems very important, particularly if applying these concepts to conservation management. If changes in species assemblages are directional, then cumulative diversity is only important in terms of legacy effects of past species. By contrast, cyclical changes in species assemblages would suggest that cumulative diversity is the "true" diversity in a given location. I would be very excited to see whether the data sets used to calculate different diversity components could also be used to distinguish directional from cyclical changes in species assemblages.

2. On a related note, the examples of "functional" consequences of cumulative diversity (ll. 50-58) seem to focus on short-term changes in local diversity, not legacy effects per se.  

3. I would like to see a more detailed discussion of the processes that might drive cumulative diversity. Fig. 3 and the associated text is a great example of how this approach can be highly informative. However, the non-environmental example given (metacommunity dynamics) is not discussed in great detail. If driven by metacommunity dynamics, would we expect some links between cumulative diversity and regional species diversity (i.e. spatial gamma diversity)? Another possible mechanism is imperfect detectability of species. If we don't detect all species in every survey, then adding more surveys through time will likely reveal more species, even if local diversity never changes. In this case, we are doing a better job of characterising local diversity but it's not really an example of cumulative diversity.

4. I was not convinced by the calculation of niche breadth with temporal data. If a rock pool is highly variable through time, then a species that stays in that pool is potentially experiencing a wide range of conditions. In addition, this definition of niche breadth ignores spatial processes, such as dispersal limitation leading to localised species occurrences. 

5. Related to the previous point, I did not understand how including rock pool abundance is accounting for rarity (ll. 165-179). The other use of abundance (to distinguish core and satellite species) seems unnecessary given that the authors explicitly study changes in evenness. Wouldn't differences in evenness capture changes in core versus satellite species?

6. I had several queries about the potential applications of this approach. In l. 265 and ll. 304-306, what implications or management strategies are the authors referring to? These need to be described in more detail. And the claims in  ll. 307-325 depend on whether diversity accumulates due to directional or cyclical changes in species assemblages (see point 1). Conservation is ideally focused on persistence of species, in which case cumulative diversity is informative only if it is due to cyclical changes in assemblages. If not, then cumulative diversity could be a misleading measure of the diversity supported by a given site.

Minor comments:

ll. 29-30: this "seemingly paradoxical result" doesn't seem paradoxical at all. It's referring to two different things (short and long term diversity).

ll. 47-49: the Galapagos example is not an example of the effects of cumulative diversity. These effects are due to high instantaneous diversity.

Methods: describe semi-partial correlation factors.

ll. 231-236: could this be due to changes in abundance (drying reduces abundances, which increases evenness)? This was clearly not captured by the abundance metric (Fig. 3b) but this might be due to an assumption of linearity in the model (i.e. the effect doesn't exist in highly abundant systems, so the overall association between abundance and evenness is nonlinear).

Appendix B: I don't know that this adds very much. None of these results seem particularly surprising, and the inferences drawn (positive vs negative associations) are not used in a very nuanced way in the main text. 

Author Response

Hammond & Kolasa – Responses to Reviewer 1

Main comments:

1. It is not clear to me whether the authors believe diversity accumulates due to directional or cyclical changes in species assemblages (or both). This distinction seems very important, particularly if applying these concepts to conservation management. If changes in species assemblages are directional, then cumulative diversity is only important in terms of legacy effects of past species. By contrast, cyclical changes in species assemblages would suggest that cumulative diversity is the "true" diversity in a given location. I would be very excited to see whether the data sets used to calculate different diversity components could also be used to distinguish directional from cyclical changes in species assemblages.

We did not have an a priori expectation of diversity accumulating in a either a directional or a cyclical fashion. We do accept, however, the reviewer’s comment that this distinction can add a valuable perspective. We added this perspective to the manuscript by:

a.     Reanalyzing the rock pool data to categorize each species in each community as either permanent (always present), cyclic (recurs intermittently), a new addition (not present initially, but added later), a lost species (initially present, but lost later) or transient (appeared and disappeared).

b.     Adding a results table (Table 1, lines 189-197) reporting the percent share of each species type in the cumulative species pool. Cyclic and transient species accounted for the vast majority of the species pool.

c.     Correlating the above percentages with alpha and beta richness values to gauge which species types contributed most to average community size and compositional turnover

d.     Incorporating these results into the Discussion to make the points that:

                      i.        environmental variation may limit cumulative diversity by eliminating cyclic species (affecting alpha richness most) and excluding new community members and transient species (affecting beta richness most), (lines 291-293) and

                     ii.        permanent and cyclic species, as the recurring diversity of a site, will be somewhat amenable to local management, while transient and cyclic species will require more of a wholly metacommunity perspective and strategy (lines 300-309).

We believe the analysis has enriched the paper, and thank the reviewer for suggesting it.

2. On a related note, the examples of "functional" consequences of cumulative diversity (ll. 50-58) seem to focus on short-term changes in local diversity, not legacy effects per se.  

We understand how our examples could be interpreted as focusing on short-term changes in diversity. However, they all refer to long-lasting functional impacts as the result of a temporary occurrence by a species. We appreciate that this might not be entirely clear without consulting the papers cited. To clarify this, we added a classical example of legacy effects – long-term changes in plant communities as a result of past deer browsing (line 51-52). Further, we emphasize in our locust and cicada example that the effects on above- and below-ground processes can occur with considerable time lags. We believe that these changes make it clearer that we are discussing the long-term, rather than instantaneous, impacts that an occurrence of a species can have on a site.

3. I would like to see a more detailed discussion of the processes that might drive cumulative diversity. Fig. 3 and the associated text is a great example of how this approach can be highly informative. However, the non-environmental example given (metacommunity dynamics) is not discussed in great detail. If driven by metacommunity dynamics, would we expect some links between cumulative diversity and regional species diversity (i.e. spatial gamma diversity)?

We are glad that the reviewer found the text for Fig. 3 illuminating. We had originally kept the discussion of metacommunity dynamics short since the related factor in our statistical model explained little variance. (Local environmental variation was adequate to explain most of cumulative diversity and its components.) We are happy, of course, to delve further into mechanism on this. As mentioned above, we added text describing contributions of permanent, cyclic and transient species and explaining their occurrence in terms of a regional species pool and metacommunity dynamics. We further cite a 2009 paper from the same rock pool system that interestingly shows that a number of metacommunity mechanisms work in concert to structure local communities (lines 300-309). We believe these additions will add a layer of insight into the metacommunity element of our study.

Another possible mechanism is imperfect detectability of species. If we don’t detect all species in every survey, then adding more surveys through time will likely reveal more species, even if local diversity never changes. In this case, we are doing a better job of characterising local diversity but it’s not really an example of cumulative diversity.

We thank the reviewer for raising the issue of imperfect detectability since we neglected to mention it. It is a valid mechanism that explains how cumulative diversity can rise solely because more samples are taken. So, it is plausible that it inflates our cumulative diversity values. However, it should not impact our main results which are regressions of cumulative diversity (and its components) against environmental and non-environmental variables. This is because the diversity of each pool is characterized by the same number of samples and there is no reason to believe that the ability to detect species is different from one pool to the next. Thus, any error in estimating cumulative diversity will be the same for each pool and would not show up as systematic bias in a regression. To broach the issue and its potential impact, we added to the following text to the Discussion:

“We note that cumulative richness and diversity can also increase when a greater sampling effort over time captures difficult-to-detect species [17,18]. But this sampling effect should not distort our regression results since each pool has the same number of samples with, presumably, the same sampling error.” (lines 315-318)

4. I was not convinced by the calculation of niche breadth with temporal data. If a rock pool is highly variable through time, then a species that stays in that pool is potentially experiencing a wide range of conditions. In addition, this definition of niche breadth ignores spatial processes, such as dispersal limitation leading to localised species occurrences. 

All definitions and measures of niche breadth have their shortcomings. We calculated Levin’s niche breadth based on the spatial distributions of species which exploits a link between tolerance and distribution that is documented in dozens of papers across multitude of taxa. It is, of course, possible for a highly-tolerant species to be localized in the landscape. More generally, though, it will be a widespread habitat generalist. It is this habitat generalization/specialization that we index with Levin’s niche breadth. We note that the approach also has the advantage of reflecting tolerance not just to measured environmental parameters but to the full range of measured and unmeasured environmental parameters that a species is exposed to. Again, this makes the metric be best described as a habitat specialization, which we have emphasized in the text.

It is true that the spatial distribution of a species can be constrained by dispersal limitation, and is a shortcoming of this index. However, this should not be the case in our study for two reasons. First, an unpublished experiment in the rock pools showed that when all organisms were removed from 20 experimental pools, communities formed through dispersal within months and within a year could not be distinguished from the undisturbed communities that preceded them. Second, we aggregate data over 14 years to calculate the index and dispersal limitation is perhaps less likely to be seen over this timescale. So, while niche breadth can be calculated in various ways, our method is supported by a low likelihood of dispersal limitation and a robust link between habitat generalization and spatial distribution.

5. Related to the previous point, I did not understand how including rock pool abundance is accounting for rarity (ll. 165-179).

We are aware that the term rare can be interpreted in various way and seek to clarify its use here. We use the term rare in the sense of a species having low numerical abundance in a rock pool. In our regression models, we included the mean abundance of species in a rock pool as a covariate for habitat specialization. This is necessary because spatial distribution, upon which our index of habitat specialization, is often negatively correlated with abundance. This correlation arises from the abundance-occupancy relationship in which a species with few individuals is usually found at few sites and so appears specialized even though it may just be have low numerical abundance. Using abundance as a covariate removes this effect. Including this term thus controls for abundance or its inverse rarity which would otherwise confound estimates of habitat specialization.

The other use of abundance (to distinguish core and satellite species) seems unnecessary given that the authors explicitly study changes in evenness. Wouldn't differences in evenness capture changes in core versus satellite species?

The reviewer may be correct that the presence of occasional (satellite) species might be detectable from evenness patterns when numbers of low abundance species add to a high abundance core. But using mean abundance of species in a rock pool for this purpose has a distinct advantage in our case: It can be used as an independent explanatory variable to estimate how much cumulative diversity is explained by metacommunity dynamics. If we were to use evenness in the same way, we would be explaining cumulative diversity in terms of evenness. Since evenness is itself a component of cumulative diversity, this set-up would introduce a circularity of inferences. Mean abundance, on the other hand, provides a signature of metacommunity dynamics that is more independent of the response variables. Moreover, as a signature it is based on a simple principle of average abundance being reduced as more and more low abundance species are added, and it is backed up by simulations.

6. I had several queries about the potential applications of this approach. In l. 265 and ll. 304-306, what implications or management strategies are the authors referring to? These need to be described in more detail.

We appreciate the need for specifics in this section. To address this deficiency, we added the following text that touches on how different alpha and beta components of cumulative diversity might be managed:

“Findings, in turn, imply that different management strategies may be required for managing the various richness and evenness components of diversity over larger timescales. Alpha richness and evenness, for instance, may be well-served by existing management techniques that seek to maintain community size and structure. But heed must also be given to strategies that enable compositional and dominance turnover, such as the maintenance of cyclic environmental regimes.” (lines 329-333)

And the claims in  ll. 307-325 depend on whether diversity accumulates due to directional or cyclical changes in species assemblages (see point 1). Conservation is ideally focused on persistence of species, in which case cumulative diversity is informative only if it is due to cyclical changes in assemblages. If not, then cumulative diversity could be a misleading measure of the diversity supported by a given site.

We took this comment into consideration when addressing Point 1. The new text describes how a substantial share of cumulative diversity is permanent or cyclic, and discusses the conservation implications of that. Directional and transient change is also discussed for a more balanced and targeted section.

Minor comments:

ll. 29-30: this "seemingly paradoxical result" doesn't seem paradoxical at all. It's referring to two different things (short and long term diversity).

We may have misjudged how the short/longterm diversity distinction might appear to readers. We have therefore removed “seemingly paradoxical” from our wording.

ll. 47-49: the Galapagos example is not an example of the effects of cumulative diversity. These effects are due to high instantaneous diversity.

We thank the reviewer for pointing out this inconsistency. We have removed that sentence and replaced it with another example, as follows:

“ High biodiversity value in some freshwater ponds, for example, emerges over time from a continual turnover of species [5].” (line 47-48)

Methods: describe semi-partial correlation factors.

We have added the following sentence to the Methods section to explain our use of semi-partial correlations:

“ We assessed the importance of individual factors in multiple regressions using their semi-partial correlations which estimate their explanatory power [36].” (line 184-185)

ll. 231-236: could this be due to changes in abundance (drying reduces abundances, which increases evenness)? This was clearly not captured by the abundance metric (Fig. 3b) but this might be due to an assumption of linearity in the model (i.e. the effect doesn't exist in highly abundant systems, so the overall association between abundance and evenness is nonlinear).

We thank the reviewer for an interesting idea. The mechanism of increasing evenness with lowered abundance of the most abundant species is appealing. As they state, however, the abundance factor does not support it. Moreover, plots of abundance and evenness do not show the nonlinear relationship that would mask the mechanism.

Appendix B: I don't know that this adds very much. None of these results seem particularly surprising, and the inferences drawn (positive vs negative associations) are not used in a very nuanced way in the main text.

We can understand that Appendix B is neither surprising nor particularly exciting. Indeed, they were not intended to be. Rather, we included it to support our interpretation of mean abundance as a factor and indicator of a particular type of metacommunity dynamic. It serves as a reference for any reader who might want to see support for using the factor in this way. The simulations in the Appendix do illustrate a relatively simple concept. But we would rather keep the Appendix in the name of full documentation and disclosure for any readers who might wish to see it.

Reviewer 2 Report

This is a well-written exploration into the temporal accumulation of local diversity. This is an understudied matter as pointed out by the authors and certainly merits further attention. My only minor comments are that references 40 & 46 are not complete.

Author Response

Hammond & Kolasa – Responses to Reviewer 2

Main comments:

My only minor comments are that references 40 & 46 are not complete.

We thank the reviewer for noticing these reference errors. We have corrected them.